# Spatial-Temporal Features Based Sensor Network Partition in Dam Safety Monitoring System

**DOI:** 10.3390/s20092517

**Published:** 2020-04-29

**Authors:** Hao Chen, Yingchi Mao, Longbao Wang, Hai Qi

**Affiliations:** 1College of Water Conservancy and Hydropower Engineering, Hohai University, Nanjing 211100, China; chenhao.hnlcj@foxmail.com; 2Centre of Research and Development, Huaneng Lancang River Hydropower Co., Ltd., Kunming 650214, China; 3School of Computer and Information, Hohai University, Nanjing 211100, China; wlb@hhu.edu.cn (L.W.); qihai@hhu.edu.cn (H.Q.)

**Keywords:** network partition, spatial-temporal feature, auto-encoder, dam safety monitoring, sensor networks

## Abstract

Many various types of sensors have been installed to monitor the deformation and stress in the dam structure. It is difficult to directly evaluate the operation status of the dam structure based on the massive monitoring data. The sensor network is divided into multiple regions according to the design specifications, simulation data, and engineering experiences. The local results from sub-regions are integrated to achieve overall evaluation. However, it ignores the spatial distribution of sensors and the variation of time series, which cannot meet the real-time evaluation for the dam safety monitoring. If the network partitions can provide the preliminary foundation for analyzing the dynamic change laws of the dam’s working conditions in a real-way, we should consider the similarity of structure and stresses in the local region of the dam and the correlation among the monitoring data. A time-series denoising autoencoder (TSDA) is proposed to represent the spatial and temporal features of the nodes by compressing high-dimensional monitoring data. Then, a network partitioning algorithm (NPA) based on spatial-temporal features based on the TSDA is presented. The NPA ensures that the partition results can support the analysis of the physical change laws by introducing the auxiliary objective variable to optimize the network partition objective function. Experimental results on the public datasets and a real dataset from an arch dam demonstrate that the proposed network partition algorithm NPA can achieve better partition performance than TSDA+K-Means and TSDA+GMM. The NPA can improve the silhouette coefficient by 45.1% and 58.4% higher than the TSDA+K-Means and TSDA+GMM, respectively. The NPA can increase the Calinski-Harabaz Index by 30.8% and 61.6%, respectively.

## 1. Introduction

With the rapid development of Internet of Things (IoT) technologies, various types of IoT devices (e.g., sensor nodes) can be deployed in large dam engineering to measure the different physical quantities and sense their changes in various regions of the structure, such as deformation, stress, pressure, etc. [1,2,3,4]. In the dam safety monitoring systems, the massive monitoring data are generated from the deployed sensor nodes. However, it is difficult to process a large amount of data with the traditional mechanical model, so it is impossible to directly evaluate the safe operation states of the dam. In real applications, the monitoring network is usually divided into multiple regions according to the design specifications, simulation data, and engineering experience. The local evaluation results of each region are integrated to achieve the global overall evaluation [5]. Taking an arch dam as an example, based on the dam sections and elevation, the monitoring network can be divided into multiple grids adopting horizontal and vertical partitions, respectively. Figure 1a shows the gridded partition results of the dam safety monitoring network based on the mechanical models. Different color solid dots correspond to different types of sensor nodes, and the horizontal and vertical dotted lines indicate the dam sections and elevation, respectively. Due to instrument failure, environmental changes, and monitoring requirements, the spatial distribution of sensor nodes vary [6]. Meanwhile, during the dam operation period, the measured data and their temporal features dynamically change with time.

Monitoring data analysis adopts the idea of “divide and conquer”, which is an important algorithm based on multi-branched recursion. A divide and conquer algorithm works by recursively breaking down a problem into two or more sub-problems of the same or related types, until these sub-problems become simple enough to be solved directly. The solutions to the sub-problems are then combined to give a solution to the original problem. To evaluate the whole structure of the dam based on the monitoring data, the monitoring network of the dam can be divided into multiple regions based on the spatial-temporal features of the monitoring data from which can be obtained the local evaluation result in the regions. The local evaluation results are combined to obtain comprehensive evaluation results. Therefore, how to divide the monitoring network becomes the important problem. In this paper, the network partition problem based on the spatial-temporal features of the monitoring data is equal to the clustering problem. The clustering problem was extensively studied in various applications, e.g., image processing, feature selection, and data stream clustering. In the image features encoding, the whole image can be divided into individual blocks with the average region division method [7]. For feature selection, an unsupervised feature selection based on subspace clustering represents samples to the subspace [8], and we can obtain the data distribution through the subspace clustering. To improve the accuracy of monitoring data processing and analysis in the sensor networks, a hierarchical aggregation clustering protocol is proposed [9]. The detected events are classified in the local sensor nodes and are transmitted to the sink node. The sink node constructs the aggregation clustering tree to obtain the event classification results. In the wireless sensor networks, the network topology is divided into several regions based on the nodes’ attributes and correlations in order to reduce energy consumption and balance the workload [10]. Considering the redundant deployment of nodes, the backbone nodes in the differential pattern tree are used to execute the task of events classification, which can reduce the computation cost and communication overheads [11]. Thus, the cluster head nodes can aggregate the monitoring data and make the decision quickly. In the applications of data stream clustering, considering the data distribution in each dimension, the subspace clustering method of the data stream based on the region partition was proposed to divide the data dimension based on the unit cell. Furthermore, it adopted a bottom-up strategy to gather the final clustering results [12]. Moreover, in order to decrease the congestion in the large-scale networks, the networks were partitioned into regional sub-networks. Some clustering methods for network partition were discussed [13]. However, the mentioned clustering methods did not consider spatial and temporal features.

At present, the arch beam load sharing methods are adopted to deploy the sensor nodes for the arch dam safety monitoring. The sensor network can be partitioned into the spatial arch beam monitoring grids based on the beam sections and arch monitoring base. The evaluation of each monitoring part is limited to the arch beam grid from the original section and base. For a monitoring part under the influence of internal and external loads of the dam, there is a correlation between the changing trends of different monitoring items, such as temperature, stress, deformation, osmolality, as well as the correlation of the operation states among the various parts of the arch dam after the completion of arch sealing. Unfortunately, we lack quantitative and feasible evaluation methods of correlation analysis. In addition, it is impossible to accurately simulate and predict the load and structural changes of the arch dam through the structural mechanics calculations during the design stage. Therefore, the arch beam grids are difficult to effectively reflect the dynamic change law of the physical quantities of arch dams, under the conditions of inter-annual cycle changes and long-term accumulation of various loads. The monitoring data of the arch dam varies dynamically and the corresponding data features also change with time. The monitoring network partition based on the arch beam load distribution depends on the structural mechanics calculations in the design stage. It is quite difficult to establish the relationship of the stress load between the adjacent monitoring parts in the arch dam in a real-time manner. Therefore, the existing network partition methods cannot timely reflect the dynamic changes of the dam structure, which results in the low-efficient evaluation of dam operation states.

We should consider the similarity of structure and stresses in the local region of the dam and the correlation among the monitoring data. A time-series denoising autoencoder (TSDA) is proposed to compress the high-dimension monitoring data in order to extract their spatial-temporal features. Considering the correlation among the spatial positions of the different sensor nodes and the change law of the time-series, the auxiliary distribution variables are introduced to optimize the objective function of deep clustering. Based on the spatial-temporal features of monitoring data, a network partition algorithm (NPA) is presented to cluster all sensor nodes and partition the sensor network into the different regions. Figure 1b shows the partition results based on the spatial-temporal feature of sensor nodes. The solid dots surrounded by different dotted boxes correspond to different regions clustered by the NPA. Compared with the traditional monitoring network partition based on the arch beam load sharing method, the partition results with the NPA method can reflect the relationship between the stress and load between the different monitoring parts of the arch dam based on the spatial-temporal data features. Furthermore, the divided areas are bilaterally symmetrical in the horizontal direction, and the solid dots with the same color are clustered into the same area. That is to say that the partition results reflect the high correlations among the sensor nodes within the same region based on the spatial-temporal features.

The paper is organized as follows. We introduce the related work in Section 2. The preliminary and problem statement are presented in Section 3. A time-series denoising autoencoder is proposed to compress the high-dimensional data for the spatial-temporal feature representation. In Section 5, we present our proposed NPA approach for sensor network partitioning based on spatial-temporal features. Finally, we evaluate the performance of the proposed NPA with four public datasets and large-scale real dam monitoring data in Section 6, and conclude the work in Section 7.

## 2. Related Work

Traditional clustering methods include partition-based clustering, fuzzy clustering, hierarchical clustering, and density-based clustering. The most widely used algorithms are K-Means clustering, Gaussian mixture model (GMM) [14,15,16], and fuzzy cluster analysis. However, traditional clustering methods still have a long convergence time and low clustering accuracy when processing high-dimensional data. Furthermore, an autoencoder (AE) is adopted to compress high-dimensional data (astronomy, meteorology, long time series, etc.) and represent their features [17,18,19,20]. Denoising autoencoders (DAEs) have been used to learn new representations for a wide range of machine learning tasks [21,22]. A deep clustering network (DCN) can learn the complex probability distribution functions to process high-dimensional data such as images and texts, but the complex network structures and parameter adjustments result in extra computation overheads [23].

Besides the mentioned works on the clustering with region division methods, there are also several studies regarding the combination of clustering with feature embedding learning. A kernelized K-means algorithm (DisKmeans) was embedded to a low dimensional subspace via a linear discriminant analysis (LDA), which was jointly learned with K-means clustering assignments [24]. An unsupervised feature selection method was proposed to combine the clustering and feature embedding tasks to achieve better performance [25]. However, these methods suffer from the shallow and linear embedding functions. A deep structure, named TAGnet was introduced in [26], where two layers of sparse coding followed by a clustering algorithm are trained with an alternating learning approach. Similar work was presented that formulates a joint optimization framework for discriminative clustering and feature extraction using sparse coding [27,28]. However, the inference complexity of sparse coding forces the model to reduce the dimension of input data with Principal Components Analysis (PCA) or use an approximate solution. Moreover, hand-crafted features and dimension reduction techniques degrade the clustering performance by neglecting the distribution of input data. To employ deep learning in graph clustering, a simple method was presented to learn a non-linear embedding of the affinity graph using a stacked autoencoder and obtain the clusters in the embedding subspace via K-means [29]. A semi non-negative matrix factorization was extended to stacked deep semi Non-negative Matrix Factorization (NMF) in order to capture the abstract information in the top layer [30]. A recurrent-based framework, named JULE, was introduced to cluster the data. In JULE, data is represented via a convolutional neural network and embedded data is iteratively clustered using an agglomerative clustering algorithm [31]. JULE achieved good results using the joint learning approach, however, it requires tuning of massive hyper-parameters, which is not practical in real-world clustering tasks. 

## 3. Problem Statement

The monitoring data in the large dams are high-dimensional dynamic spatial-temporal data. Considering the correlation among the spatial positions of the different sensor nodes and their change rules of the time series, the network partition algorithm should cluster all sensor nodes and partition the sensor network into the different regions based on the spatial-temporal features of nodes.

Assume that there are n sensor nodes installed on a dam for the dam safety monitoring. The set of sensor nodes is X={xi|i=1,…,n}, where each node xi∈ℝdx. ℝdx represents the original features of the observed data, including the spatial features (spatial location, the type of sensor node) and temporal features (time series of data). dx is the dimension of the spatial-temporal features. Figure 2 illustrates the procedure of the network partition. In the dam safety monitoring networks, the spatial features are usually static, while the temporal features of the time series are high-dimensional and constantly changing. Using the embedding function with sparse coding, φ:X→Z, the original samples are mapped into the latent feature space. In other words, the latent feature space is Z={zi|i=1,…,n}, where each latent feature element zi∈ℝdz has a much lower dimension compared to the original input data (i.e., dz≪dx). Therefore, the function with spare coding φ:X→Z can support the dimensions reduction and features representation. Given the embedding features, we can adopt a multinomial logistic regression function (SoftMax) fΘ:Z→P to compute the probability pij of the sensor node xi belonging to the jth cluster, where Θ={θi|i=1,…,m}. The probability pij can be computed as follows:(1)pij=f(zi,Θ)=eθjTzi∑j=1meθjTzi
where Θ={θ1,…,θm}∈ℝdz×m are the SoftMax function parameters. The SoftMax function is adopted to compute the probability pij and the stochastic gradient method is used to optimize the parameters Θ={θi|i=1,…,m}.

Therefore, the sensor network partition problem is equivalent to the sensor nodes clustering problem. In other words, n sensor nodes are clustered and partitioned into m regions, and we can have a set of regions $R=\{r_{j}|j=1,\ldots ,m\}$. The probability $p_{ij}$ of sensor node $x_{i}$ belonging to the jth region can be computed in Equation (1). When, $p_{ij}=\max\{p_{ik}|k=1,\ldots ,m\}$, the sensor node $x_{i}$ is partitioned into the region $r_{j}$.

In this paper, we proposed a monitoring network partition method based on the spatial-temporal features of the monitoring data. We can obtain the clustering analysis results of sensor nodes with the proposed NPA algorithm. Under the influence of the internal and external loads of the dam, our approach can provide a feasible solution for correlation analysis of the different monitoring items (e.g., temperature, stress, deformation, and seepage pressure) in the same monitoring parts. In addition, our method can support the correlation analysis of the physical effects of the adjacent monitoring parts of the dam. This paper focuses on the correlation analysis of the monitoring data themselves. Our approach provides a preliminary foundation for the dam safety evaluation, rather than directly evaluate the structural safety of the dam.

## 4. Spatial-Temporal Feature Representation

### 4.1. Temporal Feature Representation

In general, the time-series data of the sensor nodes is a variable one-dimensional series with periodicity. In order to represent the change law of the temporal features of the sensor nodes, we select the monitoring data from more than one observation period (e.g., one year) as the sample. The monitoring data is not really always periodic. For the newly constructed dam, the monitoring data includes the time effect. After 5–10 years of stable operation, the time factor has little effect. In this paper, the arch dam has been operated for 10 years and the reservoir has multi-year regulation capacity. The monitoring data exhibits periodic change. Unlike the low-dimensional and discrete spatial data, the time-series of the sensor nodes are high-dimensional and continuously changing data. Due to the hardware, network conditions, the limitation in the memory, and computational capacity of the sensor nodes, the monitoring data of sensors are often misreported or lost. Therefore, feature compression and noise reduction of the spatial-temporal data from the sensor nodes are required.

To extract and represent the features of high-dimensional and noisy time-series data, a time- series denoising autoencoder (TSDA) is proposed in this paper, as shown in Figure 3. To enhance the anti-noise ability of the TSDA, random noise data is added to the sample data set during the training process. In the encoding phase, the convolutional layer (Conv2D) and the maximum pooling layer (MaxPooling2D) are applied to the compression representation of the time-series data features. In the decoding phase, the convolutional layer and the up-sampling layer (UpSampling2D) as opposed to the encoding phase, are used to reconstruct the compression representation into the original input. Due to the same input and output of the proposed TSDA autoencoder, its objective function is to minimize the reconstruction errors and optimize the encoder and decoder.

Assume the time series data of one sensor node $x_{i}$ with the length λ is $T_{x_{i}}=[t_{1},t_{2},\ldots ,t_{\lambda }]$, and its feature of time-series (FT) can be denoted as $FT_{x_{i}}=[ft_{1},ft_{2},\ldots ,ft_{t_{x}}]$, where $t_{x}$ is the number of dimensions and $t_{x}\ll \lambda$. The detailed procedure of TSDA is as follows.

(1)Input. To facilitate the convolution, pooling, and up-sampling operations, the incoming time- series $T_{x_{i}}=[t_{1},t_{2},\ldots ,t_{\lambda }]$ is transformed into a matrix ${T^{\prime }}_{x_{i}}$ through the reshape operation, and then the matrix ${T^{\prime }}_{x_{i}}$ is mixed Gaussian random noise to generate the input of the TSDA.(2)Encoding. Multiple convolutional layers and maximum pooling layers are stacked alternately to compress the input data and obtain the feature representation FT.(3)Decoding. Multiple convolutional layers and up-sampling layers are stacked alternately to restore the data feature representation FT into the reconstructed input.(4)Objective Function. The error of the original input and the reconstructed input is taken as a loss function. The objective is to minimize the error.

After the TSDA training is completed, the weights are saved. The feature representation FT from the encoder is the temporal feature compression representation.

### 4.2. Spatial Feature Representation

The spatial features of one sensor node include the node’s coordinates, the importance of the node, and the instrument type, which can be processed with normalization. Suppose the spatial feature of one sensor node $x_{i}$ can be denoted as $FS_{x_{i}}=[fs_{1},fs_{2},\ldots ,fs_{s_{x}}]$, where $s_{x}$ is the number of dimensions.

The spatial position of the sensor node has three attributes: longitude (e), latitude (n), and height (h). In the real measurement network systems, the value ranges of different attributes are large and the dimensional units are different. It is difficult to quantitatively describe the relative position relationship amongst the senor nodes. To eliminate the side effects of different dimensional units and the large range of values, uniform dimensions, and nonlinear normalization are adopted to calculate the relative spatial position of the sensor node in order to accelerate the convergence of training. In the coordinate system, all dimensional units are uniform in meters. To avoid negative values of attributes, the symbolic function $y=sgn(x)\frac{\ln(|x|)}{\ln(\max|x|)}$ is used for data transformation and normalization.

### 4.3. Examples

In this paper, the real dataset from the highest arch dam in the world is from 1 January 2017 to 31 December 2017 from 964 sensor nodes, which have 350,000 data items recording the sensors’ types, the spatial coordination positions, time slots, and the observed data. The observed physical quantities include stress, pressure, deformation, opening degree, seepage, and temperature, etc. Then, the normalized time-series data of one sensor can be encoded as a 1×15 feature representation vector with the TSDA method. The similarity between the temporal feature vectors of two sensor nodes can be measured with cosine similarity. For example, the sensor node P04616 deployed in the middle of the dam is selected as a reference point, then we can analyze their similarities considering the temporal and spatial features of the sensor nodes. Note: this paper focuses on the clustering analysis of dam monitoring data, so there is no introduction to the structure of a specific dam. In addition, the so-called symmetry refers to the symmetry of the layout of the installed sensors, rather than the symmetry of the dam structure. The spatial arrangement of the sensors for this arch dam is symmetric with the arch crown beam as the center.

We will exhibit three different spatial correlations of sensors, as shown in Figure 4, Figure 5 and Figure 6. The arch dam is a thin shell structure. A small number of sensors are installed in the *Y*-axis direction. In order to simplify the representation, the sensors installed in the *X*-axis and *Z*-axis directions are selected as the analysis objects.

(1)Asymmetrical spatial position and long-distance. Three vertical measuring nodes P04616, P00002, and P07981 are selected to monitor the radial deformation of the arch dam. The spatial distribution and the sequential process lines of three nodes are shown in Figure 4. From Figure 4, three nodes measure the same monitoring items, and their spatial positions are irrelevant. The corresponding process line trends and fluctuation ranges are completely different. The cosine similarities of temporal feature encoding of both P04616 and P00002, and P07981 are −0.017 and 0.248, respectively. The sensor nodes with different spatial features have low temporal feature encoding.

(2)Spatial proximity of different nodes. Taking three nodes P04616, P04615, and P04617 as examples, they are installed in the same dam section and height. P04616 and P04615 are vertical measuring nodes. P04616 and P04615 are to monitor the radial displacement and tangential displacement, respectively. P04617 is the stress measuring node to monitor the radial stress of the dam. The reason for selection is to illustrate that nodes monitoring different items installed in the same part of the dam may have similar and consistent trends. As shown in Figure 5, although two nodes P04616 and P04617 monitor the different physical quantities, their observed values are similar and have basically consistent trends. Their cosine similarity of feature encoding is 0.991. On the contrary, the sequential process line of P04615 is significantly different from that of P04616, and the cosine similarity is 0.266. Thus, even with two sensor nodes with close spatial distance, the similarity of the temporal feature is also low.

(3)Symmetrical spatial positions. Taking three nodes P04616, P02023, and P06656 as examples, two nodes P02023 and P06656 are symmetrical with node P04616 as the center in the vertical direction, as shown in Figure 6a. P04616 is the vertical measuring node to monitor the radial displacement. P02023 and P06656 are the static level nodes to monitor the vertical displacement of the dam. The reason for selection is to show that two nodes monitoring the same item, which are installed at the 1/4 arch crown beam at both ends of the left and right banks of the centerline, also have high similarities of time-series data. In Figure 6b, the sequential process lines of P02023 and P06656 change in a similar trend, and the corresponding cosine similarity is 0.831. The cosine similarity between P06656 and P04616 is −0.134, and the feature difference is obvious. The reason is that the symmetric spatial structures exist with similar stress in the local regions, e.g., the dam sections on both sides bear similar pressure. There are strong similarities in the temporal features of two sensor nodes that are symmetrical in spatial positions.

From Figure 4 to Figure 6, the spatial correlation of sensors is different, and the correlation of time-series characteristics of the measured values are also different. If the sensor network partition only considers the temporal features of the nodes, the sensor nodes scattered spatial positions with similar temporal features are likely clustered into the same region, such as node P02023 and P06656. Meanwhile, if we only consider the spatial features of the nodes, the nodes with different temporal features are likely partitioned into the same region, such as P04616, P04615, and P04617. Therefore, the proposed TSDA extracts and represents the spatial-temporal features of the nodes, it can obtain reasonable network partition results and realize a high degree of cohesion in a single region.

## 5. Network Partition Based on Spatial-Temporal Features

### 5.1. Partition Objective

Given a certain sensor node $x_{i}$ in the monitoring network systems, the time-series denoising autoencoder (TSDA) can represent the temporal features of data as $FT_{x_{i}}$, and the spatial feature after normalization can be represented as $FS_{x_{i}}$. Using the feature mapping function with sparse coding $\varphi (x_{i})=\{FT_{x_{i}},FS_{x_{i}}\}=z_{i}$, the raw samples are mapped into the latent feature space.

Suppose that $p_{ij}$ is the probability of sensor node $x_{i}$ belonging to the jth cluster and P is the probability distribution of model predictions, $q_{ij}$ is the target probability and Q is the target probability distribution. The network partition problem can be transformed into the parameter optimization problem. Expectation maximization (EM) is adopted to make the prediction distribution P as close to the real probability distribution Q. In order to define the clustering objective function, we use an auxiliary target variable (that is the target probability distribution Q) to refine the model predictions probability P iteratively.

Firstly, Kullback-Leibler (*KL*) divergence [30] is introduced to decrease the distance between the model prediction P and the target variable Q.
(2)$$ℒ=KL(Q∥P)=\frac{1}{n}\overset{n}{\underset{i=1}{\sum }}\overset{m}{\underset{j=1}{\sum }}q_{ij}\log\frac{q_{ij}}{p_{ij}}$$

The degenerate solution allocates most of the samples to a few clusters or assigns a cluster to the outlier sample. In order to avoid the degenerate solution, a regularization term is imposed on the target variable Q. Therefore, we define the empirical label distribution of target variables Q as:(3)$$\Gamma_{j}=\frac{1}{n}\sum_{i=1}^{n}q_{ij}$$
where $\Gamma_{j}$ can be considered as the soft frequency of cluster assignments in the target distribution. Adopting this empirical distribution Γ, we can ensure our preference for having balanced assignments by adding the following *KL* divergence to the loss function.

Assume H is the uniform prior to the empirical label distribution Γ. If there is any extra knowledge about the distribution of network partition (clustering), we can use an arbitrary distribution to initialize the uniform prior H. The objective function ℒ is as follows:(4)$$ℒ=KL(Q∥P)+KL(\Gamma ∥H)=[\frac{1}{n}\sum_{i=1}^{n}\sum_{j=1}^{m}q_{ij}\log\frac{q_{ij}}{p_{ij}}]+[\frac{1}{n}\sum_{j=1}^{m}\Gamma_{j}\log\frac{\Gamma_{j}}{H_{j}}]=\frac{1}{n}\sum_{i=1}^{n}\sum_{j=1}^{m}q_{ij}\log\frac{q_{ij}}{p_{ij}}+q_{ij}\log\frac{\Gamma_{j}}{H_{j}}$$
where the first term $\frac{1}{n}\sum_{i=1}^{n}\sum_{j=1}^{m}q_{ij}\log\frac{q_{ij}}{p_{ij}}$ minimizes the distance between the target and model prediction distributions, the second term $q_{ij}\log\frac{\Gamma_{j}}{H_{j}}$ balances the frequency of clusters in the target variables. Utilizing the balanced target variables, we can ensure the model has more balanced predictions (cluster assignments) P directly. It is also simple to change the prior from the uniform distribution to any arbitrary distribution in the objective function ℒ if there is any extra knowledge about the probability distribution of clusters.

### 5.2. Iterative Solution to the Partition Problem

The objective function ℒ of the network partition (clustering) contains hidden variables (target variables Q, empirical distribution Γ, and the prior distribution H). The EM method is utilized to optimize the objective function. The iterative solution includes two main phases: (1) target variables Q estimation via fixed parameters; (2) parameters update via fixed target variables Q. 

(1) target variables Q estimation via fixed parameters

Using the alternative learning approach, we can estimate the target variables Q via fixed parameters and update the parameters while the target variables Q are assumed to be known (maximization step). The problem to infer the target variable Q has the following objective:(5)$$\underset{Q}{\min}=\frac{1}{n}\sum_{i=1}^{n}\sum_{j=1}^{m}q_{ij}\log\frac{q_{ij}}{p_{ij}}+q_{ij}\log\frac{\varphi_{j}}{h_{j}}$$
where the target variables are constrained to $\sum_{j=1}^{m}q_{ij}=1$, m is the number of clusters. All sensor nodes are clustered into their own regions through the network partition algorithm; thus, we have $\sum_{j=1}^{m}p_{ij}=1$. This problem can be solved using the first order method, which only requires the objective function value and its gradient at each iteration. We use the partial derivative of the objective function with respect to the target variables as follows:(6)$$\frac{\partial ℒ}{\partial q_{ij}}=\frac{q_{ij}}{\sum_{i=1}^{n}q_{ij}}+\log\left(\frac{q_{ij}\Gamma_{j}}{p_{ij}}\right)+1$$

There are a large number of sensor nodes deployed in the monitoring systems when the number of sensor nodes n is often big enough, we have $\underset{n\to \infty }{\lim}\left(q_{ij}/\sum_{i^{\prime }=1}^{n}q_{i^{\prime }j}\right)=0$. To investigate the problem more carefully, the gradient can be approximately computed by removing the first item in Equation (7). Setting the gradient equal to zero, we compute the closed-form solution and update the target variables Q.
(7)$$q_{ij}=\psi (p_{ij},\Theta )=\frac{p_{ij}{\left(\sum_{i^{\prime }=1}^{n}p_{i^{\prime }j}\right)}^{-\frac{1}{2}}}{\sum_{j^{\prime }=1}^{m}p_{ij^{\prime }}{\left(\sum_{i^{\prime }=1}^{n}p_{i^{\prime }j^{\prime }}\right)}^{-\frac{1}{2}}}$$

(2) parameters update via fixed target variables Q

When the target variables are known, we update the network parameters Θ using the estimated target variables with the following objective function.
(8)$$\underset{\Theta }{\min}=-\frac{1}{n}\sum_{i=1}^{n}\overset{m}{\underset{j=1}{\sum }}q_{ij}\log p_{ij}$$

To minimize the parameters Θ of the soft max layer in the deep neural network, the objective function can be converted into a standard cross-entropy loss function for clustering. The parameters of the soft max layer Θ can be updated by backpropagating the error. The above two steps are executed alternately during the iteration until convergence.

### 5.3. Network Partitioning Algorithm

To obtain reasonable network partition results in the dam safety monitoring networks, one network partition algorithm based on the TSDA (NPA) is presented to cluster all sensor nodes and partition the network into the different regions based on the spatial-temporal features. In order to establish the relationship between the stress and load among the adjacent dam space elements, the network is partitioned into regions based on the spatial-temporal of the monitoring data. 

Assume those n sensor nodes deployed in the monitoring network are clustered into m regions. The set of sensor nodes X={xi|i=1,…,n} is the input, the distribution of model predictions P={pij|i=1,…,n;j=1,…,m} is the output. The NPA algorithm is designed for the unsupervised clustering task and does not require a set number of regions m, however, the NPA needs to manually set the hyperparameters including training batches, maximum iterations, and iteration error thresholds. Algorithm 1 shows a brief description of the NPA algorithm.
**Algorithm 1.** Network Partition Algorithm based on Spatial-Temporal Features (NPA)**Input**: The set of sensor nodes X={xi|i=1,…,n};**Output:** The probability distribution of model prediction P={pij|i=1,…,n;j=1,…,m};1: Using the embedding function φ:X→Z, and the original samples are mapped into the latent feature space2: K-Means cluster algorithm is used to initialize the objective distribution Q
3: **While not** convergence4: fix the objective distribution Q to compute pij=f(zi,Θ)
5: update the prediction distribution P
6: fix the parameters Θ to compute $q_{ij}=\psi (p_{ij},\Theta )$
7: update the objective distribution Q
8: minimize $\underset{\Theta }{min}-\frac{1}{n}\sum_{i=1}^{n}\sum_{j=1}^{m}q_{ij}\log p_{ij}$ through fixing the objective distribution Q
9: update the parameters Θ
10: **End while**11: **Return**
$P=\{p_{ij}|i=1,\ldots ,n;j=1,\ldots ,m\}$

The proposed TSDA uses the embedding function with sparse coding φ:X→Z for the temporal feature compressed representation. The uniform dimension and nonlinear normalization are adopted to represent the spatial feature. The original samples can be mapped into the latent feature space. To avoid the uncertainty of the random initialization Q, K-Means, or GMM is selected to initialize the objective distribution Q. In order to speed up the convergence of objective distribution, EM is used to make the prediction distribution P close to the real probability distribution Q. An auxiliary target variable (that is the target probability distribution Q) is introduced to refine the model predictions probability P iteratively. The iterative solution includes two main phases: (1) target variables Q estimation via fixed parameters; (2) parameters update via fixed target variables Q.

## 6. Performance Evaluation

### 6.1. Experimental Setup

#### 6.1.1. Datasets

In this paper, the network partition method NPA based on the spatial and temporal features of data was presented. In the section of the problem statement, the sensor network partition problem is equivalent to the sensor nodes clustering problem. In order to show that the NPA works well with various datasets, we have chosen the real dam safety monitoring dataset and public datasets. The real dataset from the highest arch dam in the world is the same as the dataset mentioned in Section 4.3. The public datasets come from the Keras Framework [32] and the UCI Machine Learning Repository [33], which are widely selected to evaluate the clustering performance. Table 1 provides a brief description of each dataset.

#### 6.1.2. Baselines

We evaluate the proposed NPA in comparison with state-of-the-art clustering methods on the above datasets. K-Means clustering and the Gaussian mixture model (GMM) are widely used clustering methods. The autoencoder (AE) is usually adopted to compress the high-dimensional data and represent their features. Therefore, K-Means and GMM are added to the AE for improvement as a benchmark method in the experiments, respectively. On the public datasets, AE+K-Means, AE+GMM, and the deep clustering network (DNN) are used for comparison, meanwhile, the experimental results of K-Means and GMM are retained. On the dam monitoring datasets, K-Means and GMM are further improved into TSDA+K-Means and TSDA+GMM by using the time-series denoising autoencoder (TSDA) to extract features.

#### 6.1.3. Evaluation Metrics

Experiments on public datasets: We use three popular evaluation metrics for clustering algorithms, accuracy (ACC), normalized mutual information (NMI), and adjusted Rand index (ARI). ACC measures the extent to which the data are properly classified into the corresponding clusters, and NMI represents that the clustering results contain the amount of real information, and ARI can reflect the accuracy and purity of the clustering results. The value range of clustering accuracy (ACC) and the normalized mutual information is [0, 1]. The larger the value, the higher the clustering accuracy. The value range of the adjusted Rand index is [−1, 1], where negative values indicate poor clustering performance, and 1 indicates optimal clustering performance.

Experiments on real datasets: Three methods, TSDA +K-Means, TSDA+GMM, and the NPA, are performed to conduct experiments on the dam’s monitoring datasets. The silhouette coefficient (SC) and Calinski-Harabaz Index (CH) are used to evaluate the network partitioning performance. 

The SC quantifies the distribution of the sensor nodes. SC∈[−1,0) represents the basic error of the network partitioning performance, 0 represents the local optimal solution. If SC∈(0,1], the higher the value, the more reasonable distribution of the sensor nodes. The CH measures the degree of local cohesiveness. The higher the value, the higher the degree of cohesiveness.

### 6.2. Experimental Results on Public Datasets

We compare our NPA partition algorithm, with several baselines and state-of-the-art algorithms, including K-Means, AE+K-Means, GMM, AE+GMM, on the public datasets MNIST, Fashion-MNIST, STL-10, and Reuters. The evaluation metrics are clustering accuracy (ACC), normalized mutual information (NMI), and the adjusted Rand index (ARI). Note that the labeled data are used to tune the number of hyperparameters for each algorithm. This number only shows the number of hyperparameters, which are set differently for various datasets for more efficient performance. The experimental results are an average of 10 times. The details are shown in Table 2.

As shown in Table 2, the clustering accuracy (ACC) of K-Means on MNIST, Fashion-MNIST, CIFAR-10, and Reuters are 0.732, 0.587, 0.489, and 0.521, respectively. The ACC values of AE+K-Means are 0.887, 0.613, 0.557, and 0.762, respectively. The autoencoder (AE) can improve the clustering accuracy of K-Means by 21.2%, 4.4%, 13.9%, and 46.3%, respectively. Overall, AE+K-Means outperforms K-Means. GMM on the same four public datasets can achieve the clustering accuracy ACC 0.716, 0.495, 0.472, and 0.534, respectively, while the accuracy of AE+GMM are 0.863, 0.601, 0.523, and 0.638, respectively. The AE can increase the accuracy of GMM by 20.5%, 21.4%, 10.8%, and 19.5%, respectively. Deep clustering network (DCN) is compared with AE+K-Means and AE+GMM using the autoencoder on four datasets. DCN is inferior to the other three algorithms on the whole performance. On the Fashion-MNIST dataset, DCN can only exhibit similar performance with AG+GMM. Autoencoder can significantly improve clustering performance. AE+K-Means and AE+GMM algorithms employ autoencoder structures and use the reconstruction loss function as a data-dependent regularization for training the parameters, which can avoid undesirable local minima during training. Therefore, AE+K-Means and AE+GMM can outperform K-Means and GMM on clustering performance, respectively. MNIST, Fashion-MNIST, CIFAR-10, and Reuters are high-dimensional data. AE has the ability to effectively compress high-dimensional feature data and avoid various errors in the iteration and accelerate the convergence process.

On the MNIST dataset, clustering accuracy ACC of the NPA, AE+K-Means, and AE+GMM are 0.877, 0.863, and 0.910, respectively; NMI of the three algorithms are 0.812, 0.827, and 0.876; the ARI of the three algorithms are 0.791, 0.762, and 0.837, respectively. The NPA, AE+K-Means, and AE+GMM algorithms all adopt autoencoder for feature compression representation to obtain efficient clustering performance. 

Furthermore, we compare the NPA, AE+K-Means, and AE+GMM algorithms on the four public datasets, as shown in Table 3. The NPA can achieve better clustering performance than AE+K-Means and AE+GMM in the terms of ACC, NMI, and the ARI on the four public datasets. Especially on the CIFAR-10 dataset, the NPA can improve the clustering accuracy by about 50%, compared with AE+K-Means and AE+GMM. In general, the NPA is superior to AE+K-Means and AE+GMM in three aspects: clustering accuracy, clustering real information amount, and cluster purity.

### 6.3. Experimental Results on Real Datasets

We evaluated the proposed NPA network partition algorithm based on the real dam safety monitoring dataset, as mentioned in Section 4.3. The layout of the sensor nodes is shown in Figure 7, where the solid dots represent the sensor nodes. As shown in Figure 8, the sensor nodes are evenly deployed at the top of the dam and the number is relatively small, while the distribution of sensor nodes on the abutments is sparse and asymmetrical due to the different stresses on both abutments of the dam. The vertical dashed lines divide the dam into multiple sections. The sensor nodes are densely distributed along the dam section joints. From both sides to the middle of the dam foundation, the distribution of the sensor nodes changes from sparse to dense. The stress of the dam heel plays a key role in the overall structural safety of the dam. The sensor nodes are densely installed at the dam heel. In addition, the so-called symmetry in this paper refers to the symmetry of the layout of the monitoring sensors, rather than the symmetry of the dam structure. The spatial arrangement of monitoring sensors for the arch dam is symmetrical with the arch crown beam dam section as the center.

According to the design specifications of dam safety monitoring, the sensor nodes installed in the dam foundation and heel within the same dam section are dense. Due to the strong correlation among the sensor nodes at the same dam section, the cohesiveness of the network partition results should be as high as possible. On the other hand, the deployment of sensor nodes at the top of the dam and on both sides of the dam abutments are scattered, the range of the partitioned region is quite large. Besides, the overall structure of the dam is symmetrical, and the network partition of dam safety monitoring should also exhibit a symmetrical effect.

The proposed NPA adopts the TSDA to extract the temporal features of the sensor nodes. To evaluate the partition results of the proposed NPA algorithm, we compare the NPA with the modified AE+K-Means (TSDA+K-Means) and modified AE+GMM (TSDA+GMM), as illustrated in Figure 8. The NPA is the unsupervised clustering algorithm, and there is no need to set the number of regions m in advance. TSDA+K-Means and TSDA+GMM need to set the number of regions. The number of regions m is estimated to be 51 and 53 using the canopy and affinity propagation clustering algorithm, respectively, while the experts estimate the number of regions to be 48. In the experiments, the median is taken, so m=51.

As shown in Figure 8, the different colors of sensor nodes indicate different partitioned regions and the sensor nodes in the same region have the same color. Overall, the partitioned regions with TSDA+NPA exhibit the symmetrical distribution at the center of the dam. The sensor nodes in the same dam section, foundation, and heel are highly cohesive, while the nodes located at the top of the dam and both sides of the dam abutments are relatively scattered, and the corresponding regions are large. The TSDA+NPA can obtain reasonable partitioning results. On the contrary, the partitioning results with TSDA+K-Means and TSDA+GMM are unsatisfactory. The sensor nodes in the same dam section and dam heel are often partitioned into different regions with poor cohesiveness. The partition results of sensor nodes with TSDA+K-Means and TSDA+GMM are unreasonable.

Two metrics are introduced to evaluate the partition performance of the dam safety monitoring network. They are the silhouette coefficient (SC) and the Calinski-Harabaz Index (CH). The SC can quantify the distribution of sensor nodes. The higher the value, the more reasonable distribution. The CH can measure the degree of cohesiveness of network partitioning. The higher the value, the higher the degree of cohesiveness. Table 4 shows the average SC and CH values of three algorithms. The SC and CH of the NPA are significantly higher than those of TSDA+K-Means and TSDA+GMM. The SC of the three algorithms are all in the range of (0, 1]. Compared with TSDA+K-Means and TSDA+GMM, the SC of the NPA increases to 45.1% and 58.4%, respectively. As to the CH, the NPA can increase by 30.8% and 61.6%, respectively. The cohesiveness of the sensor nodes in the same region is high and the correlation between different regions is low. The NPA exhibits better performance on the network partitioning than the two other algorithms. The experimental results in Table 4 are consistent with those shown in Figure 7.

## 7. Conclusions

In order to describe the dynamics of the physical quantities for the dam safety monitoring through the network partition in a real-time way, a network partitioning algorithm based on the spatial-temporal features of sensor nodes (NPA) was proposed. A time-series denoising autoencoder is presented to compress the high-dimension data in order to extract the spatial-temporal features. Adopting the TSDA model, the NPA introduces an auxiliary objective variable to optimize the partition objective functions to cluster the sensor nodes, so that the partition results of the sensor network can reflect the physical laws of the dam safety operation effectively. The experimental results on the public datasets show that the proposed NPA can improve the partition accuracy and cohesiveness compared with AE+K-Means, AE+GMM, and DCN in the terms of ACC, NMI, and the ARI. For the real dataset from the dam safety monitoring system, the experimental results illustrate that the NPA can also outperform TSDA+K-Means and TSDA+GMM in the terms of network partition accuracy, the SC, and the partition distribution CH. This paper focuses on the correlation analysis of the monitoring data themselves. Our approach can provide a preliminary foundation for dam safety evaluation in a real-time way, rather than directly evaluate the structural safety of the dam. In future work, we will consider the dam safety evaluation integration of artificial intelligence and domain knowledge from the dam structure.

## Figures and Tables

**Figure 1 sensors-20-02517-f001:**
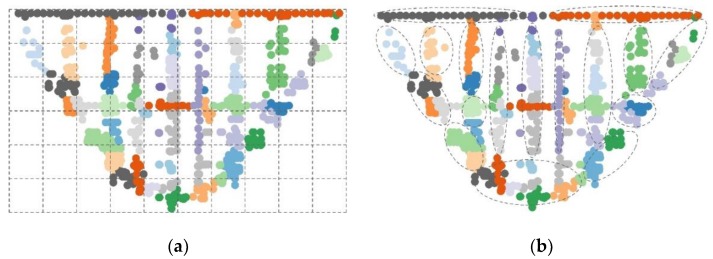
Network partition results for dam safety monitoring system (**a**) gridded partition results with a mechanical model; (**b**) partition results with the network partitioning algorithm (NPA) based on spatial-temporal features.

**Figure 2 sensors-20-02517-f002:**
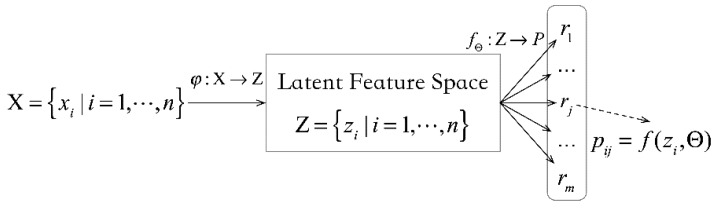
The procedure of network partitioning.

**Figure 3 sensors-20-02517-f003:**
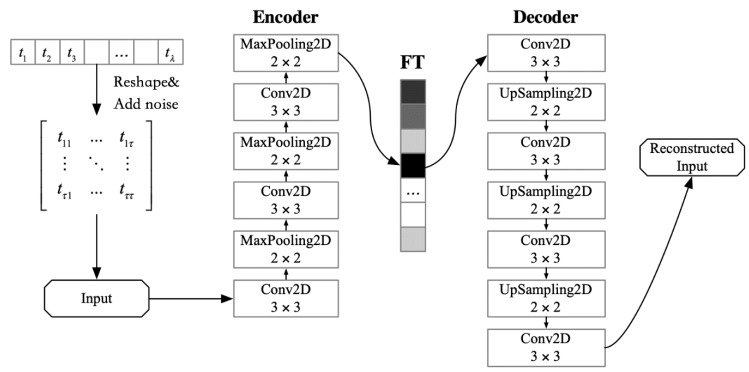
The structure of time-series denoising autoencoder (TSDA).

**Figure 4 sensors-20-02517-f004:**
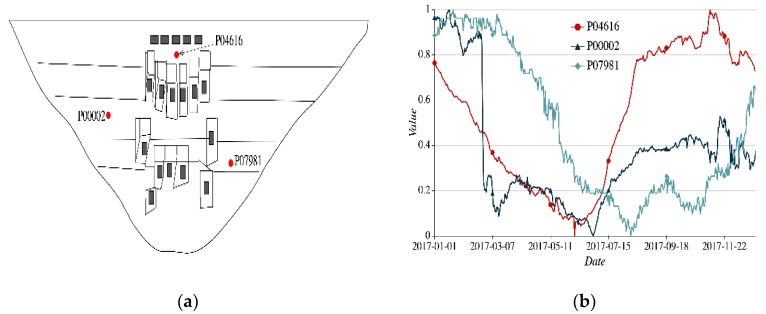
Spatial-temporal feature of nodes P04616, P07981, and P00002 (**a**) spatial distribution of three nodes; (**b**) sequential process lines of three nodes.

**Figure 5 sensors-20-02517-f005:**
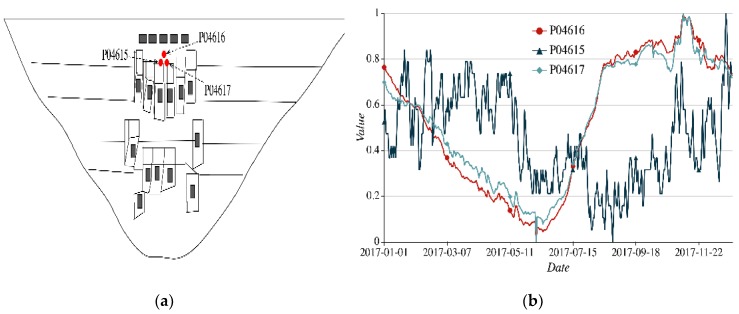
Spatial-temporal feature of nodes P04616, P04615, and P04617 (**a**) spatial distribution of three nodes; (**b**) sequential process lines of three nodes.

**Figure 6 sensors-20-02517-f006:**
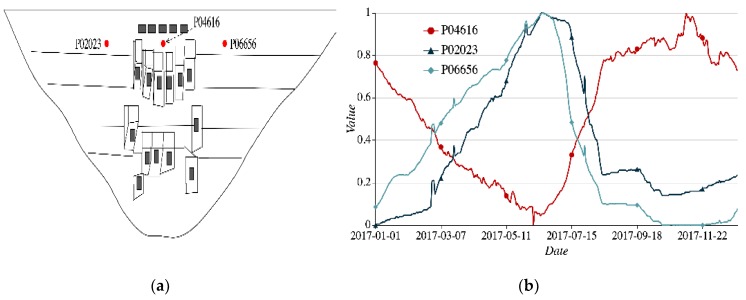
Spatial-temporal feature of nodes P04616, P02023, and P06656 (**a**) spatial distribution of three nodes; (**b**) sequential process lines of three nodes.

**Figure 7 sensors-20-02517-f007:**
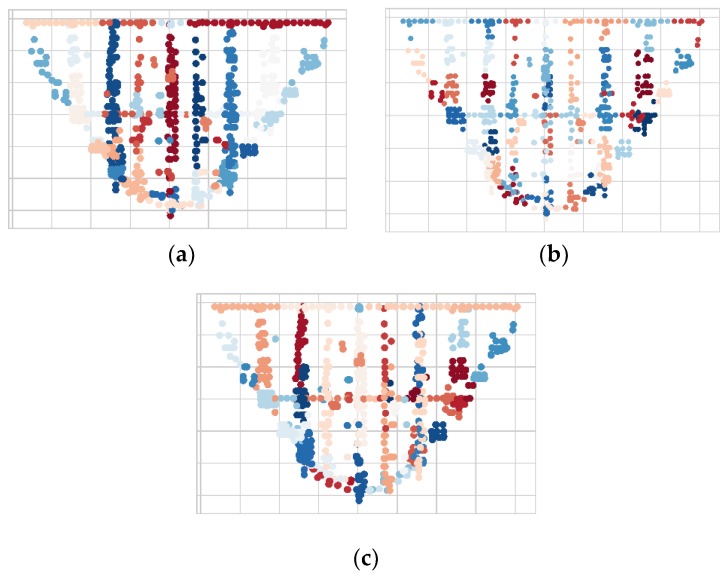
Network partition results for dam safety monitoring system (**a**) NPA; (**b**) TSDA+K-Means; (**c**) TSDA+ Gaussian mixture model (GMM).

**Figure 8 sensors-20-02517-f008:**
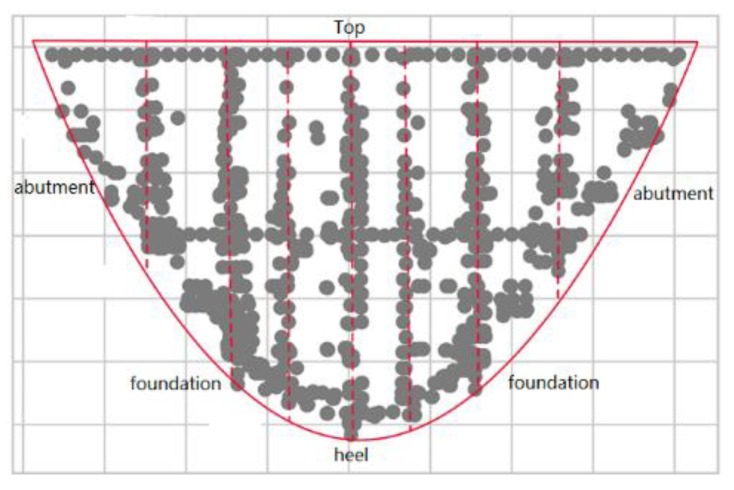
The spatial distribution of sensor nodes in the dam safety monitoring system.

**Table 1 sensors-20-02517-t001:** Public Dataset Descriptions.

Dataset	Samples	Dimensions	Classes	Description
MNIST	70,000	1 × 28 × 28	10	handwritten digits images
Fashion-MNIST	70,000	1 × 28 × 28	10	fashion items images
CIFAR-10	60,000	3 × 32 × 32	10	colorful images
Reuters	11,228	\	46	News texts

**Table 2 sensors-20-02517-t002:** Clustering performance of different algorithms on public datasets on clustering accuracy (ACC), normalized mutual information (NMI) and adjusted Rand index (ARI).

Dataset	Metrics	K-Means	AE+K-Means	GMM	AE+GMM	DCN	NPA
**MNIST**	ACC	0.732	0.887	0.716	0.863	0.813	0.910
	NMI	0.627	0.812	0.637	0.827	0.729	0.876
	ARI	0.541	0.791	0.537	0.762	0.614	0.837
**Fashion-MNIST**	ACC	0.587	0.613	0.495	0.601	0.590	0.635
	NMI	0.567	0.607	0.494	0.582	0.576	0.652
	ARI	0.405	0.524	0.321	0.478	0.453	0.563
**CIFAR-10**	ACC	0.489	0.557	0.472	0.523	0.521	0.832
	NMI	0.432	0.531	0.462	0.505	0.492	0.778
	ARI	0.378	0.470	0.417	0.489	0.436	0.691
**Reuters**	ACC	0.521	0.762	0.534	0.638	0.607	0.782
	NMI	0.312	0.537	0.371	0.482	0.446	0.674
	ARI	0.218	0.438	0.339	0.443	0.386	0.591

**Table 3 sensors-20-02517-t003:** Clustering performance of different algorithms on public datasets.

Dataset	Metrics	NPA vs. AE+K-Means	NPA vs. AE+GMM
**MNIST**	ACC	2.6%	5.4%
	NMI	7.9%	5.9%
	ARI	5.8%	9.8%
**Fashion-MNIST**	ACC	3.6%	5.7%
	NMI	7.4%	12.0%
	ARI	7.4%	17.8%
**CIFAR-10**	ACC	49.4%	59.1%
	NMI	46.5%	54.1%
	ARI	47.0%	41.3%
**Reuters**	ACC	2.6%	22.6%
	NMI	25.5%	39.8%
	ARI	34.9%	33.4%

**Table 4 sensors-20-02517-t004:** Silhouette coefficient (SC) and Calinski-Harabaz Index (CH) with TSDA+K-Means, TSDA+GMM, and the NPA.

Metric	TSDA+K-Means	TSDA+GMM	NPA
SC	0.502	0.548	0.795
CH	580.7	717.1	938.3

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
