# Peer review of "Spatial-Temporal Features Based Sensor Network Partition in Dam Safety Monitoring System"

_sensors, 2020, doi:10.3390/s20092517_

Round 1

Reviewer 1 Report

The paper presents a new approach to analyse large quantities of data obtained on dam monitoring system. The authors present algorithms and structure of the analysis procedure. After reading the paper I have the following comments and questions.

English language should be improved, and typing errors should be corrected. For example: Line 15: words stain and huge; the first one is a typing error, and I guess it should be train, and for the word »huge« we use the term large dams instead of huge dams. There are sentences starting without a capital letter, sentences where verbs are misused. The sentence: “In real applications, divide and conquer strategy is adopted.” It is only my opinion but I do not feel that a sentence has a place in an abstract of a scientific paper. Furthermore, monitoring on large dams is established to observe specific failure modes and to observe the behaviour of the dam. For that purpose, various equipment is installed on appropriate locations. This is not a divide and conquer approach it is sound engineering judgement based on the knowledge of the mechanical system, structure properties and external factors (loads,…). Data sets are poorly or not explained at all. What is public data set? What type of sensors are installed, and where, .. In line 95 the aurhors refer to “the accuracy of monitoring” I assume you mean analysis, the accuracy of monitoring is based on the quality of the installation, maintenance, etc. In line 178 you say that “the readings generated from sensors are unreliable…” if they are unreliable why do we measure then, I agree the sensors can malfunction, the readings have noise, etc… This is why data preparation is a big issue when analysing data obtained from the sensors. The structure of the dam is not presented at all. Authors refer to the symmetry in the text but this is a vague statement since we do not know the structure. It is not necessary the dam is built completely symmetrically, it depends on the site conditions, type of foundation joint, structural joints. Line 258: “sensor nodes affect each other” How? I could understand that one variable has a strong correlation to the other, but that sensors have an effect, that’s hard to believe. Please explain cohesion and partition accuracy in the context of this paper, since they are used a lot. My major concern about the paper is what exactly is the contribution? So far just a method is explained, what is the role in monitoring of dams. The analysis have been done on a one year data set, was the model validated? How do the results help with understanding the dams’ behaviour? After reading a paper I have learned nothing about the dam, since there is no interpretation, data is also poorly described. One of the messages of the paper, at least to me, is that we do not need any interpretation and sound engineering judgement, since all can be done by artificial intelligence. This is why I strongly believe that interpretation should be added.

If the authors are willing to modify their paper I recommend it for publication.

Author Response

Dear Reviewers:

We sincerely appreciate the comments from we received the editor and the anonymous reviewers. We have carefully addressed all of the comments in the revised manuscript.

Please find the revised manuscript for our paper #sensors-720753, entitled “Spatial-Temporal Features Based Sensor Network Partition in Dam Safety Monitoring System”, by Chen Hao, Wang Longbao, Mao Yingchi, and Qi Hai.

We have made substantial revisions and believe that the manuscript is much better than its previous version. Also, please find our detailed response to comments from the reviewers, which have addressed all of the points raised in the reviews.

Should you have any questions regarding this matter, please feel free to contact me.

Thank you very much.

Sincerely yours,

Yingchi Mao

College of Computer and Information,

Hohai University,

Nanjing, China, 211100

Reviewer 2 Report

In order to make the network partition reflect the dynamic changes of dam structure in real time, this paper proposes a time series denoising autoencoder (TSDA) to compress the high-dimensional monitoring data to extract the spatiotemporal characteristics of nodes, and further proposes a network partition algorithm based on spatial-temporal features. The experimental results on the public data set and real data set of arch dam show the effectiveness of the work in this paper. It's a good job.

My main concerns are as follows:

1、The contents and references of "denoising autoencoder" and "network partitioning" should be added to the relevant work and references of the paper.

2、It is suggested that the introduction and the related work should be integrated into one part, and the content of Figure 1 (b) should be put into the performance evaluation part.

3、There are some grammatical errors and spelling errors in the paper, such as “In order to the network partitions can reflect the dynamic changes of dam structure in a real-time way, it” and “Then, a Network Partitioning Algorithm based o spatial-temporal Features (NPA) based on TSDA is presented.”. It is suggested that the whole paper be revised and improved.

Author Response

  1. The contents and references of "denoising autoencoder" and "network partitioning" should be added to the relevant work and references of the paper.

Thank you for your comments. Three references are cited and added in the revised manuscript. The reference 13 is related to the network partitioning. In order decrease the congestion in the large-scale networks, the networks were partitioned into regional sub-networks. Some clustering methods for network partition were discussed [13]. But the mentioned clustering method did not consider the spatial and temporal features of data. The reference 19 and 20 presented the denoising autoencoder to represent the features in a deep neural network. All of the cited references in the revised manuscript have been highlighted.

  1. Menendez, M., Ambühl, L., Loder, A., Zheng, N., & Axhausen, K. W. Approximative network partitioning for MFDs from stationary sensor data, Transportation Research Record: Journal of the Transportation Research Board. 2019, doi: 10.1177/0361198119843264
  2. Vincent, P., Larochelle, H., Lajoie, I., Bengio, Y., and Manzagol, P.-A. (2010). Stacked denoising autoencoders: Learning useful representations in a deep network with a local denoising criterion. J. Machine Learning Res., 2010, 11.
  3. Chen, M., Xu, Z., Winberger, K. Q., and Sha, F. (2012). Marginalized denoising autoencoders for domain adaptation. Proceedings of the 29th International Conference on Machine Learning (ICML), Edinburgh, Scotland, UK, 2012.

  1. It is suggested that the introduction and the related work should be integrated into one part, and the content of Figure 1 (b) should be put into the performance evaluation part.

Thank you for your comments. If the introduction and the related works are integrated into one part, the length of that section may be too long. So, the “Introduction” section has been revised and extended through citing and discussing the references that are close to the research domain. The “related work” section focused on the discussion about the existing clustering analysis approaches and the spatial-temporal feature representations. Please find them in the revised manuscript.

In addition, in order to clearly illustrate the difference between the gridded partition results with mechanical results and the partition results based on the spatial-temporal feature of sensor nodes, two figures (Figure 1(a) and (b)) are shown in “Introduction” section. In the performance evaluation section. Figure 8 (a), (b) and (c) show the partition results with three different approaches. Figure 8 (a) illustrates that the partitioned regions with TSDA+NPA exhibit the symmetrical distribution at the center of the dam. The sensor nodes in the same dam section, foundation and heel are highly cohesive, while the nodes located at the top of dam and both sides of dam abutments are relatively scattered, and the corresponding regions are large. The TSDA+NPA can obtain the reasonable partitioning results.

  1. There are some grammatical errors and spelling errors in the paper, such as “In order to the network partitions can reflect the dynamic changes of dam structure in a real-time way, it” and “Then, a Network Partitioning Algorithm based o spatial-temporal Features (NPA) based on TSDA is presented.”. It is suggested that the whole paper be revised and improved.

Thank you for your comments. The English expression, typesetting and format have been improved and corrected in the revised manuscript. All of the modified parts in the revised manuscript have been highlighted.

Reviewer 3 Report

The authors submitted a work titled “Spatial-Temporal Features Based Sensor Network Partition in Dam Safety Monitoring System”, where they face a problem related to the sensors aimed to monitor the deformation, stress, and stain in the huge dam structure. In this context, they proposed a Time Series Denoising Autoencoder (TSDA) in order to represent the spatial and temporal features of the nodes, by compressing high-dimensional monitoring data.

In more detail, they introduce a Network Partitioning Algorithm (NPA) based on spatial-temporal Features, which is in turn based on TSDA. They claim that the NPA ensures that the partition results can reflect the physical change laws by introducing the auxiliary objective variable to optimize the network partition objective function, reporting that the experimental results related to the adoption of the proposed approach on real-world datasets gets better partition performance than the canonical approaches at the state of the art taken into account by them.

Although the manuscript is well written both in terms of technical content and organization, I suggest to the authors a careful re-reading of it in order to fix some minor typos (e.g., “based o spatial”, “ better “ without “than” in some sentences, “and” at the end of the list of authors, and so on).

The introductory part ("Introduction" section) appears unbalanced with regards to the "Related Work" section, which should be extended by citing and discussing additional papers that are very close or directly related to the domain taken into account by the authors.

The introductory section should be used only to provide a brief overview about the considered research domain, expanding each concept by using the "Related Work" section.

In addition, the author should avoid mixing literature information with information related to the proposed approach, moving them into specific section of the manuscript (e.g., "Proposed Approach") , in order to enhance the content exposure and then the general readability of the manuscript.

The experimental approach followed by the authors has been presented in a linear and clear way, but some manuscript elements should be replaced/fixed, such as, for instance, Equation 1 (some parts of the formula are almost illegible) and Figure 3 (the right part is almost illegible).

The references are updated, but accordingly to my previous observation, the author should include and discuss additional works very close or directly related to the domain taken into account, such as:

(1) Sharma, R., Vashisht, V., & Singh, U. (2020). Soft Computing Paradigms Based Clustering in Wireless Sensor Networks: A Survey. In Advances in Data Sciences, Security and Applications (pp. 133-159). Springer, Singapore.

(2) Saia, R., Carta, S., Recupero, D. R., & Fenu, G. (2019, February). Internet of entities (IoE): A blockchain-based distributed paradigm for data exchange between wireless-based devices. In Proceedings of the 8th International Conference on Sensor Networks, Prague, Czech Republic (pp. 26-27).

(3) Guerreiro, J., Rodrigues, L., & Correia, N. (2019). Resource Allocation Model for Sensor Clouds under the Sensing as a Service Paradigm. Computers, 8(1), 18.

(4) Menendez, M., Ambühl, L., Loder, A., Zheng, N., & Axhausen, K. W. (2019). Approximative network partitioning for MFDs from stationary sensor data.

The “Conclusion” section well provide a brief but complete summary of the performed work to the readers, reporting clearly the advantages related to the proposed approach.

Apart from the minor problems previously highlighted, the weakness I found in the proposed work is the absence of details about the configuration of the state-of-the-art approaches used in order to validate the proposed one (this prevents the reproducibility of the experiments carried out by them), along with the absence of clear information about the criteria used by the authors in order to select them. The authors have to provide such information, clearly.

Author Response

1.Although the manuscript is well written both in terms of technical content and organization, I suggest to the authors a careful re-reading of it in order to fix some minor typos (e.g., “based o spatial”, “ better “ without “than” in some sentences, “and” at the end of the list of authors, and so on).

Thank you for your comments. We have carefully checked and fixed all typos in the revised manuscript. that All of the modified parts in the revised manuscript have been highlighted.

2.The introductory part ("Introduction" section) appears unbalanced with regards to the "Related Work" section, which should be extended by citing and discussing additional papers that are very close or directly related to the domain taken into account by the authors.

Thank you for your comments. The “Introduction” section has been extended through citing and discussing the additional references that are close to the domain. Please find them in Line 56-80 in the revised manuscript.

  1. The introductory section should be used only to provide a brief overview about the considered research domain, expanding each concept by using the "Related Work" section.

Thank you for your comments. In the “Introduction” section has been revised and just provided a brief overview about the research work. Some concepts about the spatial-temporal feature representation, and clustering analysis are moved to in the “Related Work” section.

  1. In addition, the author should avoid mixing literature information with information related to the proposed approach, moving them into specific section of the manuscript (e.g., "Proposed Approach"), in order to enhance the content exposure and then the general readability of the manuscript.

Thank you for your comments. We have checked and modified the mixing literature information related to the proposed approach. In the “Problem Introduction” section, the network partition problem is formulated and converted into the nodes clustering problem. In addition, some explanations about the research domain are added in the revised manuscript. All of the modified parts in the revised manuscript have been highlighted.

  1. The experimental approach followed by the authors has been presented in a linear and clear way, but some manuscript elements should be replaced/fixed, such as, for instance, Equation 1 (some parts of the formula are almost illegible) and Figure 3 (the right part is almost illegible).

Thank you for your comments. All of the equations and figures have been enlarged to the legible form in the revised manuscript.

  1. The references are updated, but accordingly to my previous observation, the author should include and discuss additional works very close or directly related to the domain taken into account, such as:

(1) Sharma, R., Vashisht, V., & Singh, U. (2020). Soft Computing Paradigms Based Clustering in Wireless Sensor Networks: A Survey. In Advances in Data Sciences, Security and Applications (pp. 133-159). Springer, Singapore.

(2) Saia, R., Carta, S., Recupero, D. R., & Fenu, G. (2019, February). Internet of entities (IoE): A blockchain-based distributed paradigm for data exchange between wireless-based devices. In Proceedings of the 8th International Conference on Sensor Networks (SENSORNETS-2019), Prague, Czech Republic (pp. 26-27).

(3) Guerreiro, J., Rodrigues, L., & Correia, N. (2019). Resource Allocation Model for Sensor Clouds under the Sensing as a Service Paradigm. Computers, 8(1), 18.

(4) Menendez, M., Ambühl, L., Loder, A., Zheng, N., & Axhausen, K. W. (2019). Approximative network partitioning for MFDs from stationary sensor data.

Thank you for your comments. The above-mentioned four references have been cited and added in the revised manuscript. Please find them in Line 39, Line 77-80. Moreover, extra two references have also been cited and added in the revised manuscript, please find them in Line 112-114.

  1. Vincent, P., Larochelle, H., Lajoie, I., Bengio, Y., and Manzagol, P.-A. (2010). Stacked denoising autoencoders: Learning useful representations in a deep network with a local denoising criterion. Machine Learning Res., 2010, 11.
  2. Chen, M., Xu, Z., Winberger, K. Q., and Sha, F. (2012). Marginalized denoising autoencoders for domain adaptation. Proceedings of the 29th International Conference on Machine Learning (ICML), Edinburgh, Scotland, UK, 2012.

7.The “Conclusion” section well provide a brief but complete summary of the performed work to the readers, reporting clearly the advantages related to the proposed approach.

Thank you for your comments. We have improve the “Conclusion” section and report the advantages of our proposed approach. This paper focuses on the correlation analysis of the monitoring data themselves. Our proposed network partitioning algorithm based on the spatial-temporal features of sensor nodes can provide the preliminary foundation for the dam safety evaluation in a real-time way. In the future work, we will consider the dam safety evaluation integration of the artificial intelligence and domain knowledge from the dam structure.

8.Apart from the minor problems previously highlighted, the weakness I found in the proposed work is the absence of details about the configuration of the state-of-the-art approaches used in order to validate the proposed one (this prevents the reproducibility of the experiments carried out by them), along with the absence of clear information about the criteria used by the authors in order to select them. The authors have to provide such information, clearly.

Thank you for your comments. In the experiments, we evaluate the proposed NPA approach in comparison with state-of-the-art approaches on the public datasets. K-Means and Gaussian Mixture Model (GMM) are widely used clustering analysis methods. The Auto-Encoder (AE) is usually adopted to compress the high-dimensional data and represent their features. Therefore, K-Means and GMM are added to the AE for improvement as a benchmark method in the experiments, respectively. On the public datasets, AE+K-Means, AE+GMM, and the Deep Clustering Network (DNN) are used for comparison, meanwhile the experimental results of K-Means and GMM are retained. On the dam monitoring datasets, K-Means and GMM are further improved into TSDA+K-Means and TSDA+GMM by using the Time Series Denoising Autoencoder (TSDA) to extract features.

On the public datasets, we use three popular evaluation metrics for clustering algorithms, Accuracy (ACC), Normalized Mutual Information (NMI), and Adjusted Rand Index (ARI). ACC measures the extent to which the data are properly classified into the corresponding clusters, and NMI represents that the clustering results contain the amount of real information, and ARI can reflect the accuracy and purity of clustering results. The value range of Clustering Accuracy (ACC) and the Normalized Mutual Information is [0,1]. The larger the value, the higher the clustering accuracy. The value range of the Adjusted Rand Index is [-1,1], where negative values indicate poor clustering performance and 1 indicates optimal clustering performance.

On the on real datasets: TSDA +K-Means, TSDA+GMM, and NPA, are performed to conduct experiments on the dam's monitoring datasets. The Silhouette Coefficient (SC) and Calinski-Harabaz Index (CH) are used to evaluate the network partitioning performance. The SC quantifies the distribution of the sensor nodes.  represents the basic error of the network partitioning performance, 0 represents the local optimal solution. If , the higher the value, the more reasonable distribution of the sensor nodes. The CH measures the degree of local cohesiveness. The higher the value, the higher the degree of cohesiveness.

Round 2

Reviewer 1 Report

Dear Authors,

I have read the corrected manuscript. This time, I added my comments directly into the pdf file, please find it eclosed.

As I already asked before please correct English. I believe the paper is interesting, I like your work, but it is very hard to read.

I wish you all the best.

Author Response

Dear Reviewers:

We sincerely appreciate the comments from we received the editor and the anonymous reviewers. We have carefully addressed all of the comments in the revised manuscript.

Please find the revised manuscript for our paper #sensors-720753, entitled “Spatial-Temporal Features Based Sensor Network Partition in Dam Safety Monitoring System”, by Chen Hao, Mao Yingchi, Wang Longbao, and Qi Hai.

We have made substantial revisions and believe that the manuscript is much better than its previous version. Also, please find our detailed response to comments from the reviewers, which have addressed all of the points raised in the reviews.

Should you have any questions regarding this matter, please feel free to contact me.

Thank you very much.

Sincerely yours,

Yingchi Mao

College of Computer and Information,

Hohai University,

Nanjing, China, 211100
